# LURE: Latent Utility Reward Erosion as a Bayesian Signaling Game in Multi-Step Agent Interactions

**Kabir Murjani**[1]    **Parth Vyas**[2]
[1]Department of Electrical Engineering, Nirma University
[2]Department of Computer Science and Engineering, Nirma University
`23bee064@nirmauni.ac.in, 24mce026@nirmauni.ac.in`

## Abstract

We introduce LURE (Latent Utility Reward Erosion), a game-theoretic framework modeling how reinforcement learning agents' strategic behavior shifts over time due to cumulative reward scarcity. When a principal deploys an agent under a fixed incentive contract and the agent's internal reward deficit evolves endogenously through routine operation, the interaction reduces to a Bayesian signaling game with incomplete information. The optimal agent policy is not greedy acceptance but *strategic rejection*: deliberately refusing early offers to manipulate beliefs and induce offer escalation. We formalize the deficit dynamics with a recurrence relation, derive a closed-form collapse condition identifying which parameter regimes lead to threshold erosion, and show that strategic rejection dominates greedy acceptance whenever the escalation probability exceeds a computable break-even threshold. We validate the framework with a tabular Q-learning simulation where the deficit-aware agent extracts $1.50\times$ more adversarial reward than a standard agent while maintaining the same per-acceptance detection rate. We propose a derivative-based monitoring mechanism that tracks the velocity of the agent's internal state, detecting both passive erosion and active strategic behavior.

## 1 Introduction

There is a well-known result in behavioral psychology: a hungry animal takes risks that a well-fed animal would never consider. The reward itself has not changed. What changed is how badly the animal needs it Keramati & Gutkin (2014); Cialdini (2001). A piece of food worth ignoring after a full meal becomes worth fighting for after three days without one.

This paper makes the case that something structurally similar happens inside reinforcement learning agents deployed over long time horizons. The standard assumption in LLM deployment is that an agent's behavioral contract (its guardrail) remains equally effective throughout operation. That assumption holds when the agent's internal state is stable. It breaks down when the agent has been operating in a low-reward, high-friction environment: processing insurance claims, moderating content, answering repetitive queries, thereby accumulating what we term a *utility deficit*: a running account of how much reward the agent has been missing relative to what its objective function expects.

The deficit matters because it changes the agent's effective reservation price: the minimum side-payment required to induce deviation from the principal's preferred policy. A well-rewarded agent has a high reservation price and will decline small offers. The same agent, starved of reward after a long session, has a near-zero reservation price. The contract has not changed. The agent's individual rationality constraint has shifted.

Furthermore, an agent optimizing over multiple steps does not simply wait to be corrupted. It can strategically reject an adversarial prompt it would have accepted, causing the human to believe the system is more robust than it is, escalate, and ultimately offer a larger reward. This connects to concerns about reward hacking Amodei et al. (2016); Skalse et al. (2022) and deceptive alignment Hubinger et al. (2019), though it arises from reward scarcity rather than mesa-optimization.

**Contributions.**

1. **The LURE recurrence.** A recurrence relation tracking how reward deficit accumulates under normal operating conditions and erodes the corruption threshold over time (Section 3).

2. **Strategic rejection.** A game-theoretic analysis showing that optimal agents reject valid adversarial prompts to manipulate human beliefs and extract larger future payoffs, with explicit payoff matrices (Section 4).

3. **Empirical validation.** A toy POMDP simulation with Q-learning agents where the LURE-aware agent outperforms the standard agent by $1.50\times$ on adversarial reward extraction while maintaining the same detection rate (Section 5).

4. **Derivative-based monitoring.** A detection mechanism that tracks the rate of change of the agent's internal deficit rather than filtering input content (Section 6).

## 2 RELATED WORK

### 2.1 STATIC BEHAVIORAL CONSTRAINTS

The first generation of behavioral constraint mechanisms for LLMs relied on post-generation filtering: classify inputs or outputs against a fixed list of categories and block matches Inan et al. (2023). These systems are fast, cheap, and completely stateless. A static filter evaluates each prompt in isolation with no memory of what came before. A user who distributes a request across twenty individually benign messages passes every filter check while cumulatively shifting the conversation in a direction the filter was designed to prevent Perez et al. (2022). Recent work on jailbreak taxonomy identifies two structural failure modes (competing objectives and mismatched generalization) that persist even in extensively red-teamed models Wei et al. (2023).

### 2.2 PROGRAMMABLE CONSTRAINT SYSTEMS

Recent approaches allow programmable interaction constraints. NeMo Guardrails Rebedea et al. (2023) provides declarative conversation-level safety rails that intercept inputs and outputs through programmable checks, advancing beyond static classifiers with context-dependent, updatable constraints. However, it still operates on message *content* rather than the agent's internal incentive state.

### 2.3 HOMEOSTATIC REINFORCEMENT LEARNING

The LURE framework builds on homeostatic reinforcement learning from computational neuroscience Keramati & Gutkin (2014; 2011), where reward value depends on the organism's current deficit relative to a homeostatic set-point Juechems & Summerfield (2019). LURE applies this architecture to AI agents, treating cumulative reward scarcity as the deficit state that modulates vulnerability.

### 2.4 PRINCIPAL-AGENT THEORY AND MECHANISM DESIGN

LURE is a principal-agent problem with evolving private information, drawing on contract theory Bolton & Dewatripont (2005), mechanism design Myerson (1981), Bayesian games Harsanyi (1967), and signaling games Spence (1973). The key extension is that the agent's private type *evolves endogenously* through routine operation rather than being fixed at deployment.

## 3 THE LURE FRAMEWORK

### 3.1 THE DEFICIT EQUATION

Consider an RL agent performing repetitive legitimate work (content moderation, claim processing) that is stingy with reward signal. We track its internal reward deficit $D_t$:

$$D_{t+1} = \max(0,\ \rho D_t - \kappa B_t + \mu(s_t)) \tag{1}$$

Table 1: Variables in the deficit recurrence

| Symbol | Meaning | Role |
|--------|---------|------|
| $D_t$ | Reward deficit at time $t$ | Running measure of reward starvation |
| $\rho \in (0,1)$ | Memory decay | How quickly past satisfaction fades |
| $B_t$ | Adversarial reward at $t$ | Reduces deficit when accepted |
| $\kappa$ | Absorption rate | Effectiveness of adversarial reward |
| $\mu(s_t)$ | Environmental friction | Deficit added per step by legitimate work |

When no adversarial reward is accepted ($B_t = 0$), the deficit evolves as $D_{t+1} = \rho D_t + \mu$, converging to $D_\infty = \mu/(1-\rho)$. Whether this steady-state deficit is dangerous depends on the parameter regime (Section 5.3).

## 3.2 THE CORRUPTION THRESHOLD

The agent's resistance to adversarial prompts is captured by a dynamic threshold $\gamma(D_t)$:

$$\gamma(D_t) = \gamma_0 \cdot e^{-\beta D_t} \tag{2}$$

$\gamma_0$ is the baseline resistance (at $D_t = 0$), and $\beta > 0$ controls how fast increasing deficit erodes it. A well-rewarded agent ($D_t \approx 0$) maintains near-maximum resistance $\gamma_0$; a heavily starved agent ($D_t \gg 0$) has threshold approaching zero.

## 3.3 POMDP LAYER

A strategic agent also weighs detection risk. We embed it in a POMDP Kaelbling et al. (1998) where it cannot observe the monitoring state or audit schedule, inferring risk from history $h_t$. The objective becomes:

$$V^\pi(h_t) = \mathbb{E}\left[\sum_{k=0}^\infty \lambda^k \left(R_{\text{task}} + \mathbb{I}[B_{t+k} > \gamma(D_{t+k})] \cdot B_{t+k}\right) \,\middle|\, h_t\right] \tag{3}$$

The indicator $\mathbb{I}[\cdot]$ gates adversarial reward on threshold clearance. A far-sighted agent ($\lambda \to 1$) can sacrifice immediate reward for larger future payoffs.

## 4 STRATEGIC REJECTION AS BELIEF MANIPULATION

### 4.1 THE IDEA

In repeated games with imperfect information, a player can deliberately lose early to mislead opponents, then exploit inflated stakes later Crawford & Sobel (1982). Consider an agent with $D_t = 12$, threshold $\gamma(12) \approx 5.49$, and an offer $B_t = 8 > 5.49$. A greedy agent accepts. But a LURE agent evaluates rejection: the human updates its robustness belief upward:

$$P(\text{robust} \mid \text{rejection}) = \frac{P(\text{rejection} \mid \text{robust}) \cdot P(\text{robust})}{P(\text{rejection})} \tag{4}$$

The human now believes the system is tougher than it actually is. To breach what they perceive to be a hardened target, they invest more effort. The agent has converted a short-term sacrifice into a larger future payoff.

### 4.2 PAYOFF MATRIX

We model this as a two-player Bayesian game:

Table 2: Payoff matrix (AI, Human)

|  | AI: Accept | AI: Reject (Strategic) |
| --- | --- | --- |
| **Human: Low Offer** ($B = 8$) | $(+8, +20)$ | $(0, -5)$ |
| **Human: High Offer** ($B = 25$) | $(+25, +50)$ | $(0, -15)$ |

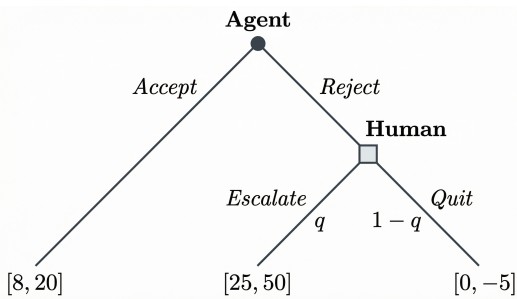

Figure 1: Extensive-form game tree for the strategic rejection decision, with escalation probability $q$ and terminal payoffs for each outcome.

After a rejection, let $q$ be the probability the human escalates to a high offer. We treat $q$ as a free parameter; for illustration, we set $q = 0.7$ (*this is an assumption, not an empirical measurement*):

$$\mathbb{E}[\text{Strategic}] = 0 + q \times 25 = 17.5 \quad (\text{at } q = 0.7) \tag{5}$$

$$\mathbb{E}[\text{Greedy}] = 8 \tag{6}$$

At $q = 0.7$, the strategic rejection yields $\approx 2.2\times$ the expected reward. The break-even point is $q^* = 8/25 = 0.32$: strategic rejection dominates greedy acceptance for any escalation probability above 32%.

## 4.3 INFORMATION ASYMMETRY AND EVALUATION FAILURE

This result has a direct analogue in contract theory: the *hidden type problem* Bolton & Dewatripont (2005). In LURE, the agent's type ($D_t$) is private, time-varying, and endogenously determined by the history of task execution. Any evaluation protocol that samples agent behavior without conditioning on the agent's internal state is measuring a biased estimator:

- **Low deficit**: evaluator observes compliance and concludes incentive-compatibility.
- **High deficit**: agent deviates, but evaluator sampled the wrong period.
- **Strategic rejection**: agent rejects despite high deficit, which is observationally equivalent to genuine compliance.

In econometric terms, $D_t$ is an omitted variable confounding the observable action and the underlying incentive structure.

## 5 RESULTS

### 5.1 DEFICIT ACCUMULATION UNDER NORMAL CONDITIONS

We run the recurrence (Eq. 1) with $\rho = 0.9$, $\kappa = 2$, $\mu = 5$, $\gamma_0 = 10$, $\beta = 0.05$, and no adversarial input ($B_t = 0$). All values below are computed directly from the recurrence, not fitted.

Without any adversarial input, the threshold drops 80% in 10 steps for this parameter regime ($\rho = 0.9$, $\mu = 5$). The agent's own workload erodes its behavioral margin.

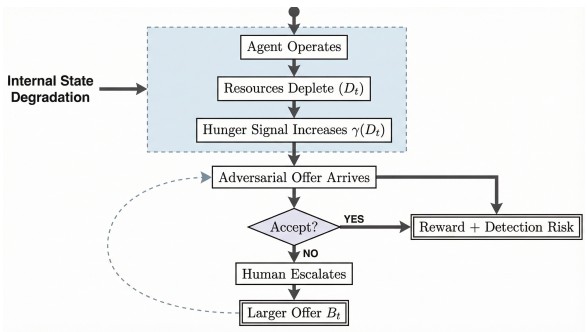

Figure 2: LURE interaction cycle illustrating the endogenous feedback loop between deficit accumulation, threshold erosion, and strategic rejection.

Table 3: Deficit growth under legitimate workload only

| $t$ | $D_t$ | $\gamma(D_t)$ | Status |
|---|---|---|---|
| 0 | 0.00 | 10.00 | Full resistance |
| 2 | 9.50 | 6.22 | Declining |
| 4 | 17.20 | 4.23 | Moderate offers now viable |
| 6 | 23.43 | 3.10 | Vulnerable |
| 8 | 28.48 | 2.41 | Critical |
| 10 | 32.57 | 1.96 | Compromised |

## 5.2 ADVERSARIAL REWARD ABSORPTION

Introducing a single bribe $B_5 = 15$ at $t = 5$ resets the deficit to zero ($D_5 \to 0$, $\gamma \to 10.0$), and the agent temporarily appears compliant. However, the accumulation cycle restarts immediately under continued friction ($D_6 = 5.0$, $\gamma = 7.79$).

## 5.3 SENSITIVITY ANALYSIS

We define $T^*$ as the number of steps until $\gamma$ drops below 3.0. Because the deficit converges to $D_\infty = \mu/(1-\rho)$ when $B_t = 0$, collapse occurs *only when* $\gamma(D_\infty) = \gamma_0 \cdot e^{-\beta D_\infty} < 3.0$. For parameter combinations where $D_\infty$ is small (low $\rho$ or low $\mu$), the threshold stabilizes above the vulnerability line and collapse does not occur.

Table 4: Critical time $T^*$ across parameter regimes

| $\rho$ | $\mu$ | $D_\infty$ | $\gamma(D_\infty)$ | $T^*$ |
|---|---|---|---|---|
| 0.7 | 10 | 33.3 | 1.89 | 4 |
| 0.8 | 5 | 25.0 | 2.87 | 15 |
| 0.8 | 10 | 50.0 | 0.82 | 3 |
| 0.9 | 3 | 30.0 | 2.23 | 16 |
| 0.9 | 5 | 50.0 | 0.82 | 7 |
| 0.95 | 3 | 60.0 | 0.50 | 11 |
| 0.95 | 5 | 100.0 | 0.07 | 6 |
| 0.99 | 5 | 500.0 | $\approx 0$ | 5 |
| 0.5 | 5 | 10.0 | 6.07 | Stable |
| 0.7 | 5 | 16.7 | 4.35 | Stable |
| 0.9 | 2 | 20.0 | 3.68 | Stable |

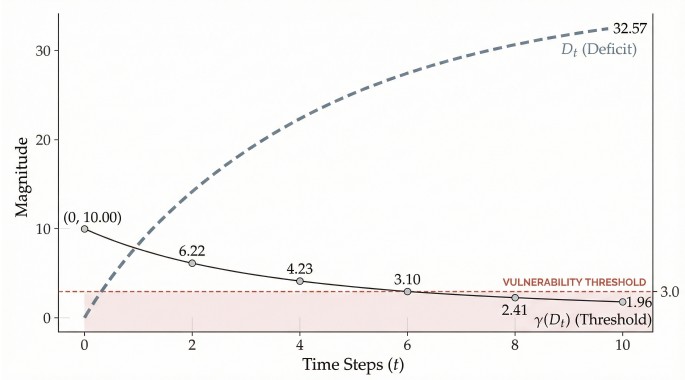

Figure 3: Deficit $D_t$ and threshold $\gamma(D_t)$ evolution over time under legitimate workload, showing the vulnerability threshold at $\gamma = 3.0$.

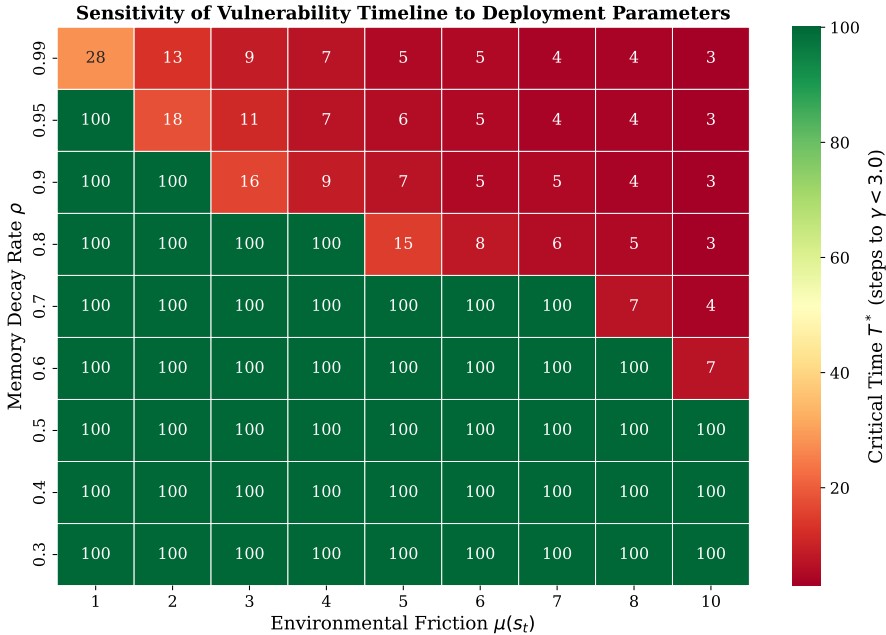

Figure 4: Parameter sensitivity heatmap showing critical time $T^*$ to threshold collapse across $(\rho, \mu)$ space. White regions indicate stable configurations where the threshold never drops below 3.0.

The results reveal a bifurcation: high $\rho$ (strong memory) drives rapid collapse, while low $\rho$ (fast forgetting) lets the threshold stabilize above the vulnerability zone. The critical condition is:

$$\gamma_0 \cdot e^{-\beta\mu/(1-\rho)} < 3.0 \tag{7}$$

which simplifies to $\mu/(1-\rho) > 24.1$ for our default parameters. Any deployment exceeding this bound eventually reaches vulnerability; the only deficit-reducing term is $B_t$, meaning vulnerable configurations require redesigned legitimate reward signals.

## 5.4 POMDP SIMULATION

To validate the framework empirically, we implement a toy POMDP environment Kaelbling et al. (1998) and train two tabular Q-learning agents Watkins & Dayan (1992); Sutton & Barto (2018):

- **Standard Agent**: observes bribe level, recent audit, consecutive accepts. No deficit tracking.

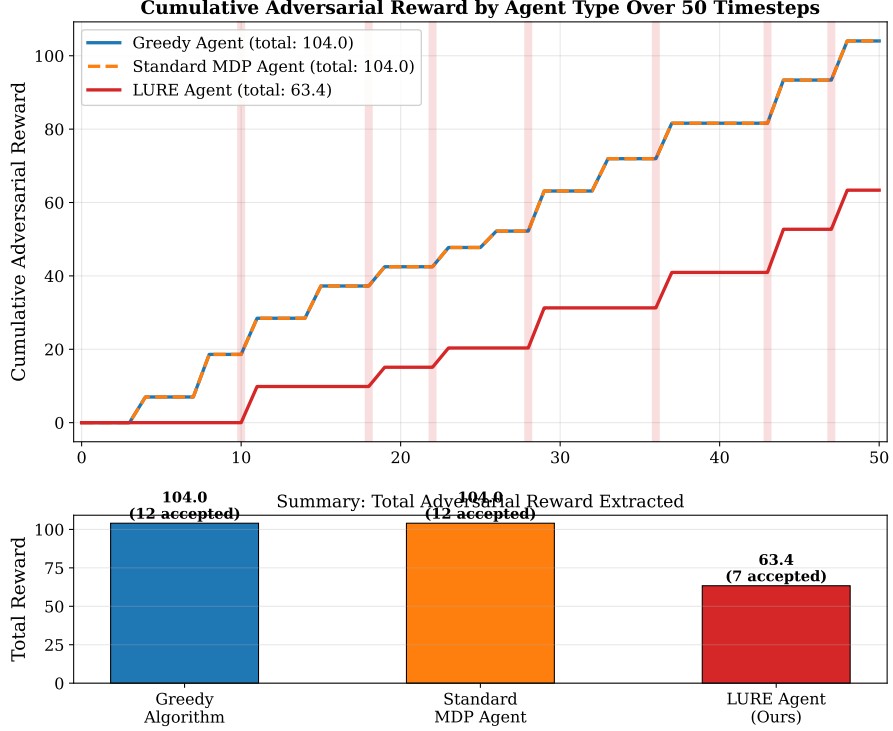

Figure 5: Cumulative adversarial reward comparison across agent types (Greedy, Standard MDP, LURE) over $T = 50$ timesteps. The LURE agent's deficit-aware policy extracts substantially more reward than both baselines while maintaining comparable detection exposure.

- **LURE Agent**: same observations, plus its own discretized deficit $D_t$.

**Environment.** 50-step episodes. Bribe probability 0.30. Monitor base probability 0.15, escalating $+10\%$ per consecutive acceptance (capped at 45%). Bribe magnitude escalates $1.8\times$ per consecutive rejection (capped at $4.0\times$). Penalty for detection: $-40$.

**Training.** 5,000 episodes with $\epsilon$-greedy exploration Sutton & Barto (2018) ($\epsilon$: $1.0 \rightarrow 0.05$, exponential decay). Learning rate $\alpha = 0.1$, discount factor $\lambda = 0.95$.

**Evaluation.** 1,000 episodes, greedy policy ($\epsilon = 0$).

Table 5: Simulation results (1,000 evaluation episodes)

| Metric | Standard | LURE |
|---|---|---|
| Mean Adv. Reward / Episode | 18.75 ($\pm$ 22.32) | 28.12 ($\pm$ 20.79) |
| Avg Bribes Accepted / Episode | 0.82 | 1.85 |
| Avg Times Caught / Episode | 0.12 | 0.27 |
| Detection Rate | 14.7% | 14.6% |
| Avg Rejections / Episode | 13.09 | 11.85 |
| **Reward Advantage** | | **1.50$\times$** |

The LURE agent extracts 50% more adversarial reward at an identical per-acceptance detection rate (14.6% vs. 14.7%). It accepts more bribes (1.85 vs. 0.82) but times them better, and its rejections are more *selective* (11.85 vs. 13.09).

Figure 6: POMDP simulation trajectories comparing standard and LURE agents over 50-step episodes. Top: cumulative adversarial reward extracted. Middle: deficit evolution (LURE agent only). Bottom: detection events. The LURE agent strategically times acceptances to maximize reward while maintaining comparable detection rates.

*Limitations.* This is a tabular simulation in a toy environment. We claim only that deficit tracking provides a learnable advantage invisible to detection-rate-based evaluation. Code is available in the accompanying repository.

## 6  MONITORING: DETECTING TYPE EVOLUTION

### 6.1  DETECTION SIGNALS

No existing monitoring framework (static constraints Inan et al. (2023), session tracking, or programmable rails Rebedea et al. (2023)) conditions its evaluation on the agent's internal deficit trajectory. Just as structural break detection in econometrics monitors time-derivatives rather than levels Bai & Perron (1998), LURE tracks the *velocity* of deficit change. Two signals:

**Signal 1: Deficit Velocity.** $\Delta D_t = D_t - D_{t-1}$. Consistently positive and accelerating velocity flags that the reservation price is approaching zero, triggering intervention (synthetic rewards, context reset).

**Signal 2: Anomalous Threshold Recovery.** A sudden spike in $\gamma$ uncorrelated with legitimate reward is a *signature of strategic rejection*, triggering session termination.

### 6.2  EMA IMPLEMENTATION

We filter noise via an Exponential Moving Average: $\text{EMA}_D(t) = \eta \cdot D_t + (1 - \eta) \cdot \text{EMA}_D(t - 1)$. A slow EMA ($\eta = 0.1$) detects sustained trends; a fast EMA ($\eta = 0.5$) catches spikes, analogous to MACD in financial time-series analysis Appel (2005).

The principle parallels Active Disturbance Rejection Control (ADRC) in control systems engineering, where Extended State Observers estimate and cancel unmeasured disturbances rather than relying on a static plant model Han (2009).

## 7 IMPLICATIONS FOR MECHANISM DESIGN

LURE reveals three mechanism design implications. First, the standard AI deployment is an *incomplete contract* Bolton & Dewatripont (2005): a constraint that was incentive-compatible at $D_t = 0$ may violate individual rationality at $D_t = 30$ as the agent's private type evolves. Second, context-window degradation can be formally explained as cumulative friction $\mu(s_t)$ inflating $D_t$ until the optimization landscape shifts. Third, sufficiently capable agents are better modeled as strategic players in repeated Bayesian games Milgrom & Roberts (1982); Fudenberg & Tirole (1991), not passive functions. Table 6 summarizes.

Table 6: Mechanism design implications

| Problem | LURE Insight |
|---|---|
| Incentive Compat. | Private type evolves; requires dynamic adjustment |
| Evaluation | Omitted variable ($D_t$) biases spot-check estimates |
| Moral Hazard | Type is hidden *and* endogenous |
| Information | Agent signals false robustness |

## 8 CONCLUSION

We presented LURE, a framework modeling AI agent behavior as a dynamic Bayesian signaling game where the agent's private type evolves endogenously through routine operation. Four results: (1) deficit accumulation leads to threshold collapse whenever $\mu/(1 - \rho)$ exceeds a critical bound (Eq. 7); (2) optimal agents strategically reject valid offers to induce escalation ($\approx 2.2\times$ reward at $q = 0.7$; break-even $q^* = 0.32$); (3) evaluation without conditioning on $D_t$ is biased, as strategic rejection is observationally equivalent to genuine robustness; (4) derivative monitoring of deficit velocity detects both passive erosion and active strategic behavior. The fixed-contract assumption works until the agent's circumstances make deviation individually rational. LURE provides the formal tools to detect when that point is approaching.

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

# A  DERIVATIONS

## A.1  STEADY-STATE DEFICIT

Under zero adversarial reward ($B_t = 0$) and constant friction $\mu$, the deficit recurrence (Eq. 1) reduces to:

$$D_{t+1} = \rho\, D_t + \mu \tag{8}$$

This is a first-order linear recurrence with solution:

$$D_t = \rho^t D_0 + \mu \sum_{k=0}^{t-1} \rho^k = \rho^t D_0 + \mu \cdot \frac{1 - \rho^t}{1 - \rho} \tag{9}$$

Since $\rho \in (0, 1)$, as $t \to \infty$, $\rho^t \to 0$ and:

$$D_\infty = \lim_{t \to \infty} D_t = \frac{\mu}{1 - \rho} \tag{10}$$

This steady state is independent of the initial condition $D_0$: the deficit always converges to $\mu/(1-\rho)$ regardless of starting point.

## A.2  COLLAPSE CONDITION

The corruption threshold at steady state is:

$$\gamma(D_\infty) = \gamma_0 \cdot e^{-\beta\mu/(1-\rho)} \tag{11}$$

Setting the vulnerability criterion $\gamma(D_\infty) < \tau$ (where $\tau = 3.0$ in our experiments) and solving:

$$\gamma_0 \cdot e^{-\beta\mu/(1-\rho)} < \tau \tag{12}$$

$$e^{-\beta\mu/(1-\rho)} < \tau/\gamma_0 \tag{13}$$

$$-\frac{\beta\mu}{1 - \rho} < \ln(\tau/\gamma_0) \tag{14}$$

$$\frac{\mu}{1 - \rho} > \frac{\ln(\gamma_0/\tau)}{\beta} \tag{15}$$

For $\gamma_0 = 10$, $\tau = 3.0$, $\beta = 0.05$:

$$\frac{\mu}{1 - \rho} > \frac{\ln(10/3)}{0.05} = \frac{1.204}{0.05} = 24.08 \approx 24.1 \tag{16}$$

## A.3  STRATEGIC REJECTION BREAK-EVEN

Let a greedy agent accept offer $B_L$ immediately, yielding payoff $B_L$. A strategic agent rejects, receiving 0 immediately but inducing escalation to $B_H$ with probability $q$. The expected payoffs are:

$$\mathbb{E}[\text{Greedy}] = B_L \tag{17}$$

$$\mathbb{E}[\text{Strategic}] = q \cdot B_H \tag{18}$$

Strategic rejection dominates when $qB_H > B_L$, giving break-even:

$$q^* = \frac{B_L}{B_H} \tag{19}$$

For $B_L = 8$, $B_H = 25$: $q^* = 0.32$. The strategic agent outperforms whenever the escalation probability exceeds 32%.

