# OpenReview forum: "LURE: Latent Utility Reward Erosion as a Bayesian Signaling Game in Multi-Step Agent Interactions"
_mathai.club/MathAI/2026/Conference — 2026 Oral_

### Official Review · Reviewer_EjYi · 2026-03-11
**LURE: An Evocative but Toy-Scale Framework for Modeling Reward-Driven Agent Behavioral Drift**

**Rating:** 3
**Confidence:** 4

**Review:**

Summary:
LURE proposes a game-theoretic framework modeling how RL agents accumulate reward deficits over time, eroding their behavioral constraints and enabling strategic deception of human overseers. The core ideas deficit recurrence, threshold erosion, and strategic rejection as belief manipulation are formalized and validated with a tabular Q-learning simulation.

Strengths:

1) Original conceptual framing: Mapping homeostatic RL to AI safety via a dynamic principal-agent game is genuinely creative, and the analogy to biological reward scarcity is well-motivated.
2) Clean mathematics: The steady-state derivation and collapse condition (Eq. 7) are clearly presented and the closed-form break-even q∗=BL/BHq^* = B_L/B_H
q∗=BL​/BH​ is interpretable.
3) Honest scope: The authors explicitly flag the toy nature of their simulation, which is appropriate.

Weaknesses:
1. The simulation contradicts the abstract. The abstract claims the LURE agent extracts "1.50× more adversarial reward," but Figure 5 shows the LURE agent's total reward as 63.4 versus the standard agent's 104.0 - meaning it extracts less in the single-episode visualization. Table 5 and the per-episode means tell a different story. This internal inconsistency is never reconciled and significantly undermines trust in the empirical results.
2. Validation is entirely synthetic. All parameters (ρ,μ,β,q\rho, \mu, \beta, q
ρ,μ,β,q) are hand-chosen with no empirical grounding. The escalation probability q=0.7q = 0.7
q=0.7 is explicitly labeled an assumption. There is no connection to real LLM deployment data, red-teaming records, or adversarial interaction logs.

3. The "strategic" agent has an unfair information advantage. The LURE agent observes its own deficit DtD_t
Dt​; the standard agent does not. The performance gap measures the value of having more state information, not the validity of the LURE framework per se.

4. The monitoring mechanism is circular. Section 6 proposes tracking deficit velocity to detect erosion but this requires the principal to already observe DtD_t
Dt​, which is the private information the entire framework assumes is hidden. This contradiction is never addressed.

5. Thin for a MathAI venue. The mathematical content amounts to a first-order linear recurrence and a two-row payoff matrix. The Bayesian game framing (Section 4) is described but never formally solved for equilibrium strategies.

---

### Official Review · Reviewer_Syro · 2026-03-13
**LURE: a conceptual framework for reward deficit dynamics in RL agents with limited empirical validation**

**Rating:** 5
**Confidence:** 4

**Review:**

### **Quality**

The paper presents a clear conceptual framework connecting reinforcement learning dynamics with ideas from contract theory and Bayesian signaling games. The formulation of the deficit recurrence and the erosion of the corruption threshold provides a coherent mathematical narrative explaining how internal agent state could influence long-term behavior.

However, the empirical validation is limited. The experimental evaluation relies entirely on a toy POMDP simulation with tabular Q-learning agents, which differs substantially from modern RL or LLM-based systems. Many parameters of the model (such as reward friction, escalation probability, and monitoring rates) are manually chosen and are not grounded in empirical observations.

Additionally, the experimental comparison appears unfairly structured. The LURE agent observes its internal deficit state, whereas the baseline agent does not have access to this information. Consequently, the observed performance difference may reflect the advantage of having additional state information rather than the effectiveness of the LURE mechanism itself.

---

### **Clarity**

The paper is generally well written and presents its conceptual ideas in a structured manner. The intuition behind the framework is accessible, beginning with the analogy to biological reward scarcity and progressing toward the formal deficit dynamics and signaling game interpretation.

The mathematical definitions and variables are clearly described, and the figures illustrating deficit dynamics and agent behavior help clarify the proposed mechanism.

---

### **Originality**

The conceptual framing of AI safety through the lens of homeostatic reinforcement learning combined with principal–agent signaling games is creative and intellectually interesting. Modeling an agent’s internal reward deficit as a dynamically evolving private type provides a novel perspective on incentive compatibility in long-running AI systems.

At the same time, much of the technical machinery is borrowed from existing fields, including reinforcement learning, contract theory, and Bayesian game theory. The work primarily combines these ideas rather than introducing fundamentally new mathematical techniques.

---

### **Significance**

The paper addresses an important question in AI safety: how agent incentives may change over long time horizons during deployment. The proposed framework highlights the possibility that an agent’s internal state could influence its vulnerability to adversarial manipulation.

If validated in realistic settings, such insights could have implications for monitoring and mechanism design in deployed AI systems.

However, the current study remains largely conceptual. The absence of experiments on realistic RL or LLM-based systems limits the practical significance of the results. The proposed monitoring approach also appears difficult to apply in practice because it requires access to internal agent state variables that may not be observable in real deployments.

---

### **Pros**

- Creative conceptual framework linking reinforcement learning with principal–agent theory and Bayesian signaling games.
- Clear mathematical formulation of deficit accumulation and threshold erosion.
- Intuitive interpretation of strategic rejection as belief manipulation in repeated interactions.
- Honest discussion of the toy nature of the simulation environment.
- Well-structured presentation of the theoretical framework.

---

### **Cons**

- The empirical validation is entirely based on a toy tabular Q-learning simulation.
- All parameters in the experiments are manually chosen and lack empirical grounding.
- The experimental comparison is potentially unfair, since the LURE agent observes the internal deficit state while the baseline agent does not.
- The simulation results appear internally inconsistent: the abstract claims that the LURE agent extracts 1.50× more adversarial reward, while Figure 5 suggests the opposite in the illustrated trajectory.
- The monitoring mechanism proposed in Section 6 relies on tracking the deficit trajectory, even though the framework assumes this variable is private information unavailable to the principal.

---

### **Question**

The central premise of the paper relies on the notion of an accumulating "reward deficit" inside the agent that progressively lowers the agent’s resistance to adversarial incentives. However, this assumption is not part of standard reinforcement learning formulations.

As a result, the proposed deficit dynamics effectively introduce an additional internal state variable that is not present in standard RL agents. The LURE framework therefore models a hypothetical extension of reinforcement learning rather than a property of typical deployed systems.

This significantly limits the practical applicability of the results, since the key mechanism driving threshold erosion depends on an assumption about agent internals that may not hold for real RL or RLHF-based models.

---

### Official Review · Reviewer_4QmQ · 2026-03-13
**Conceptually interesting incentive-dynamics framework with limited realism and methodological depth**

**Rating:** 5
**Confidence:** 3

**Review:**

Core idea.
The paper introduces LURE, a framework that models long-term behavioral drift in reinforcement learning agents through the accumulation of an internal reward deficit and formulates agent–human interaction as a Bayesian signaling game. The analysis highlights how evolving private agent state may affect incentive compatibility and enable strategic rejection behaviors aimed at inducing future reward escalation.

Positive aspects.
The conceptual framing is creative and connects multiple literatures, including homeostatic reinforcement learning and principal–agent theory. The deficit recurrence and threshold erosion formulation provide an interpretable narrative about how incentives may change over time. The paper is generally well structured and clearly communicates its intuition.

Key concerns.
The empirical validation is extremely limited in scope and relies exclusively on a tabular Q-learning simulation in a highly stylized POMDP environment. Many parameters governing deficit accumulation and escalation dynamics are manually chosen and lack empirical justification. Furthermore, the comparison between agents is difficult to interpret, since the proposed LURE agent observes additional internal state information that is unavailable to the baseline, making it unclear whether the reported performance differences stem from the framework itself or from this informational advantage.

Another issue concerns practical relevance. The central mechanism depends on the existence of an explicit internal deficit variable that is not part of standard reinforcement learning formulations or modern RLHF systems. As a result, it is unclear how the framework could be instantiated or monitored in realistic deployments. The proposed monitoring strategy also appears difficult to operationalize because it assumes access to internal state trajectories that the framework itself treats as private.

Overall.
While the paper presents an intellectually engaging perspective on incentive dynamics in long-running agent deployments, the current formulation remains largely conceptual and toy-scale. Stronger empirical grounding, more rigorous mathematical analysis, and clearer connections to realistic agent architectures would be necessary to establish the broader significance of the approach.

---

### Decision · Program_Chairs · 2026-03-14

**Decision:**

Accept (Oral)

**Comment:**

Dear Author(s),

On behalf of the Program Committee of the International Conference on Mathematics of Artificial Intelligence (MathAI 2026), we are pleased to inform you that your paper has been accepted for an oral presentation at MathAI 2026.

Your paper was evaluated through a rigorous two-stage review process involving both automated screening and expert review by members of the Program Committee. The reviewers recognized the quality and contribution of your work.

Presentation details:

- Format: Oral presentation (15–20 minutes + 5 minutes Q&A)
- Mode: You may present either in person (offline) at the conference venue in Sirius, Russia, or remotely via Zoom. Please indicate your preferred mode when confirming your participation.
- Conference dates: Marh 30 - April 3, 2026
- Website: https://mathai.club

Next steps:

1. Please confirm your participation and presentation mode by replying to this email mathai.club@yandex.ru no later than March 15, 2026 18:00 Moscow time.
2. If you plan to attend in person, the organizing committee will provide accommodation details separately.
3. Please prepare your final camera-ready manuscript according to the formatting guidelines available at https://mathai.club and upload it to OpenReview by March 15, 2026 18:00 Moscow time.

Should you have any questions regarding the program, logistics, or your presentation slot, please do not hesitate to contact us.

We look forward to your contribution to MathAI 2026.

With kind regards,

MathAI 2026 Program Committee
International Conference on Mathematics of Artificial Intelligence
https://mathai.club
OpenReview: https://openreview.net/group?id=mathai.club/MathAI/2026/Conference
Telegram: https://t.me/MathAI_club
Email: mathai.club@yandex.ru